# Detection of Chikungunya virus in bodily fluids: The INOVACHIK cohort study

**Ezequias B. Martins**[1]\*, **Michele F. B. Silva**[1], **Wagner S. Tassinari**[2], **Fernanda de Bruycker-Nogueira**[3], **Isabella C. V. Moraes**[1], **Cintia D. S. Rodrigues**[3], **Carolina C. Santos**[3], **Simone A. Sampaio**[3], **Anielle Pina-Costa**[1], **Allison A. Fabri**[3], **Vinícius Guerra-Campos**[3], **Nayara A. Santos**[1], **Nieli R. C. Faria**[3], **Ana Maria B. Filippis**[3], **Patrícia Brasil**[1], **Guilherme A. Calvet**[1]

**1** Acute Febrile Illnesses Laboratory, Evandro Chagas National Institute of Infectious Diseases, Oswaldo Cruz Foundation, Rio de Janeiro, Rio de Janeiro, Brazil, **2** Mathematics Department, Exact Sciences Institute, Federal Rural University of Rio de Janeiro, Rio de Janeiro, Rio de Janeiro, Brazil, **3** Flavivirus Laboratory, Oswaldo Cruz Institute, Oswaldo Cruz Foundation, Rio de Janeiro, Rio de Janeiro, Brazil

\* ezequias.martins@ini.fiocruz.br

**Data Availability Statement:** All relevant data are within the manuscript.

**Funding:** This work was supported by grants from INOVA-Fiocruz (Grant VPPCB-008-FIO-18-223) to

## Abstract

### Background

Chikungunya is a widely distributed, re-emerging tropical disease caused by the chikungunya virus (CHIKV). Little is known about the duration for which CHIK RNA are detectable in bodily fluids, especially genital secretions, and current evidence is based on small series or case reports. An understanding of viral dynamics across different body compartments can inform diagnostic testing algorithms and public health prevention interventions.

### Methodology

A prospective cohort study was conducted to assess the presence and duration of detectable levels of CHIKV RNA in blood, urine, saliva, semen, and vaginal secretions. Men and women (≥ 18 years) with a positive reverse transcriptase-polymerase chain reaction (RT-PCR) test for CHIKV in the acute phase (1–14 days) of the disease were included. After enrollment, clinical data and samples were collected every 15 days over the first 2 months, and a final collection was performed 3 months after recruitment. The Kaplan–Meier interval-censoring method and the parametric Weibull model were fitted to estimate the median time of viral persistence until the lack of CHIKV RNA detection among all body fluids. Punctual estimates of the median time of CHIKV RNA persistence for each fluid were estimated using a 95% confidence interval (CI).

### Results

From April to December 2019, 170 participants were screened. Of these, 152 (100 women) were enrolled in the study. The median and interquartile range (IQR) ages for men and women were 39.3 (IQR: 26.9, 50.7) and 43.5 (IQR: 33.8, 53.6) years, respectively. CHIKV RNA was detected in 80.3% (122/152) of serum samples, 23.0% (35/152) of urine samples, 30.3% (46/152) of saliva samples, 14.3% (6/42) of semen samples, and 20.2% (20/99) of vaginal secretion samples. The median time until the loss of CHIKV RNA detection was

GAC and the Flavivirus Laboratory was supported by Coordenação de Vigilância em Saúde e Laboratórios de Referência / CVSRL /Fiocruz, by Fundação de Amparo à Pesquisa do Estado do Rio de Janeiro/ Faperj under the grant no. E26/ 2002.930/2016 and by the Horizon 2020 ZIKACTION under the Grant 734857. The funders had no role in the design of the study; in the collection, analyses, or interpretation of data; in the writing of the manuscript, or in the decision to publish the results.

**Competing interests:** The authors have declared that no competing interests exist.

19.6 days (95% CI, 17.5–21.7) in serum, 25.3 days (95% CI, 17.8–32.8) in urine, 23.1 days (95% CI, 17.9–28.4) in saliva, and 25.8 days (95% CI, 20.6–31.1) in vaginal secretion. The number of semen samples available was too small to make statistical estimates, but a last positive sample was obtained from a participant 56 days after the onset of symptoms.

## Conclusions

CHIKV RNA could be detected in all bodily fluids studied, including genital secretions during the acute and convalescent phases and additional studies on viral infectivity in semen and vaginal secretions are warranted.

### Author summary

This prospective cohort study of adult patients aimed to estimate the presence and duration of detectable levels of chikungunya virus RNA in bodily fluids, including genital secretions, among participants in the acute and convalescent phases of the disease. In addition to the fluids usually used for diagnosis in humans (serum and plasma), we reported the detection of chikungunya virus RNA in all body fluids. Reports have shown that chikungunya virus RNA in serum declines to undetectable levels within 1–2 weeks after symptom onset. The viral persistence in the serum in our study was longer than expected. In addition, we showed that saliva and urine contained detectable viral RNA in both the acute and convalescent phases of the disease. To the best of our knowledge, this is the first cohort study assessing the presence and persistence of CHIKV in genital fluids (vaginal secretions and semen). Knowledge of viral persistence can help inform recommendations for the control, treatment, and prevention of the disease. Additional studies on viral infectivity are warranted.

## Introduction

Chikungunya is a widely distributed, re-emerging tropical disease caused by the chikungunya virus (CHIKV) and transmitted by *Aedes aegypti* mosquitoes [1,2]. Prior to 2013, CHIKV cases and some outbreaks were identified in several countries in Africa, Asia, Europe, and the Indian Ocean Islands. In 2013, the first local transmission of the CHIKV in the Americas was identified in the Caribbean countries and territories. The virus then spread throughout most of the Americas in 2014, especially in Brazil [1,2]. To date, three CHIKV genotypes are known: West African, Asian, and East Central South African (ECSA), of which the last two are prevalent in Brazil [3,4].

Clinically, the disease has three main phases: acute, post-acute, and chronic. In the acute phase, symptom onset ranges from 2–12 days following bites by an infected mosquito. This phase is associated with an abrupt onset of fever, headache, arthralgia, myalgia, fatigue, prostration, and rash. Severe joint pain is the most prevalent symptom, described in 90% of cases. The post-acute phase appears after 14 days of illness, following the febrile period. At this stage, joint pain is observed, which may last for up to 3 months. Finally, in the chronic phase, articular manifestations are seen, which are usually debilitating and can persist for many years [5,6].

Routinely, diagnosis is performed using serum or plasma samples [7]. However, the alternative use of urine and saliva samples for molecular diagnosis has been described in the acute

phase of flavivirus infections, such as the West Nile virus [8], dengue virus (DENV) [9], and Zika virus (ZIKV) infections [10]. In addition, several studies have suggested a more extended detection and persistence of ZIKV in selected body fluids, such as saliva, urine, semen, sweat, and rectal samples [11–15].

Gardner et al. showed that oral fluid (saliva) of CHIKV-infected animals and humans might contain infective CHIKV in the acute phase of the disease. Human saliva samples were obtained from 13 CHIKV-positive patients who presented with hemorrhagic manifestations [16]. The prolonged detection of ZIKV RNA in semen has been described in some studies [13,17,18]. Although studies involving the isolation of CHIKV in saliva, urine, and semen samples are scarce, during an outbreak of chikungunya in French Polynesia, the virus was detected in saliva and urine samples in the acute phase of the disease [19]. Bandeira et al. reported CHIKV RNA in semen and urine samples 30 days after symptom onset, bringing new perspectives for alternative diagnostic forms and mechanisms of infection transmission [20], with implications for its prevention and control.

This study aimed to estimate the presence and duration of detectable levels of CHIKV RNA in bodily fluids, namely serum, saliva, urine, semen, and vaginal secretion in the acute and convalescent phases of the disease.

## Methods

### Ethics statement

INOVACHIK was a prospective cohort study conducted at the Acute Febrile Illness Laboratory, Oswaldo Cruz Foundation outpatient clinic in Rio de Janeiro, Brazil. The institutional review board reviewed and approved the study protocol (CAAE: 06779019.0.0000.5262). Written informed consent wasobtained before participation from all patients.

### Study site and cases management

Patients admitted to the hospital or the intensive care unit were not targeted for enrollment to avoid bias towards patients with more severe disease. Thus, the patients enrolled in the study were screened at a general febrile illness outpatient clinic for more generalizable findings. Patients seen at this outpatient clinic are either referred by other health units in Rio de Janeiro or spontaneously seek care.

Men and women aged ≥18 years who had developed acute fever or arthralgia (with or without a rash) and no evident focus of bacterial infection within the previous 7 days were enrolled. A standard case report form was used to record information about the epidemiological and clinical features. We defined fever as an axillary temperature ≥ 37.5˚C. Patients with symptoms reported for up to 7 days were included in the study. The first visit (with fluid collection) was performed on different days, depending on the patient's arrival at our clinic.

Clinical data and biological samples were collected every 15 days for 2 months, with a final 3-month collection. The first samples for all fluids were collected in the first week of symptoms, and the second samples were collected within 14 days after the onset of symptoms (+/- 3 days as visit window). Data regarding clinical signs and symptoms was collected during the acute phase (1–14 days). All patients were tested over the study duration (3 months), despite undetectable RT-PCR results in all body fluids collected during a visit.

### Laboratory tests

Serum, urine, saliva, semen, and vaginal secretion specimens were collected at enrollment, every 15 days for 2 months, and at the 3-months follow-up.

The samples were tested for CHIKV using real-time reverse transcriptase polymerase chain reaction (rRT-PCR). Following the manufacturer's instructions, RNA was extracted using the QIAmp Viral RNA Mini Kit. The general procedures for rRT-PCR for chikungunya, zika, and dengue have been described elsewhere [21,22,23]. The RT-PCR mix was prepared using the GoTaq Probe 1-Step RT-qPCR System and was run using the Applied Biosystems 7500 Real-Time PCR System. Cycle threshold (Ct) values lower than 38 and sigmoid curves were considered positive. In addition, serum was tested for anti-CHIKV-IgM according to the manufacturer's protocol.

As a reference laboratory, all measures to avoid cross-contamination within the samples were adopted. There were different areas designated only for PCR: 1. In two large rooms, RNA extraction (manual or automated) was performed in cabinets with UV light or in a biological safety cabinet, with an adjacent room for adding samples in the PCR mix; 2. One "clean" room was used for mix preparation only; 3. One room with thermocyclers was used for the amplifications.

A non-template control (NTC) was used to check for the absence of sample cross-contamination, contamination of reagents, consumables, and environment. The NTC is a negative "sample" that can be water or negative plasma extracted simultaneously with the clinical samples and included in the amplification process. In all steps, tips with barriers were used, most consumables were disposable, and gloves were changed frequently. In addition, aseptic cleaning was frequently performed in all rooms.

Testing for ZIKV and DENV by rRT-PCR, was also performed for serum samples collected during the acute phase of the disease using a commercial kit (ZDC) from the *Instituto de Tecnologia em Imunobiológicos Biomanguinhos*. The ZDC kit was approved by the *Agência Nacional de Vigilância Sanitária/ANVISA* (registry #80142170032). Urine specimens collected during the acute phase were also tested for ZIKV using rRT-PCR. In cases where the ZDC Kit detected DENV, the protocol by Lanciotti *et al.* was used to identify DENV subtypes [24]. All study tests were performed at the National Reference Laboratory for Epidemiological Surveillance of Arbovirus in the Laboratory of Flavivirus at the Oswaldo Cruz Institute, Fiocruz, Rio de Janeiro, Brazil.

## Statistical analyses

The sociodemographic variables were described using frequencies and proportions for categorical variables and medians and ranges or interquartile ranges (IQRs) for continuous variables [25].

An exploratory analysis was performed by calculating summary statistical measures, and the violin plot was used to assess the distribution of the persistence time of the fluids. The Kaplan–Meier (K–M) curve technique was used to analyze time-to-event outcomes and estimate the probability of survival at various time intervals. Graphs were used to illustrate survival; in this case, the persistence time for each fluid over time [26]. The time until the lack of RNA detection in each bodily fluid was defined as the number of days between the onset of CHIKV symptoms and the first negative RT-PCR result. We assumed that CHIKV RNA in all specimens was detectable on the day of symptom onset. Patients who never tested positive during the study were excluded from the analysis, even if they had only one visit. Parametric Weibull regression models were used to estimate the time until the loss of CHIKV RNA detection in body fluids. Medians and 95% confidence intervals (CIs) were used to report the results. We estimated survival functions and 95% CIs for the Weibull model with median and 95$^{th}$ percentiles [27].

The Weibull curve was used as a smoother curve for the Kaplan–Meier estimator's distribution. The Weibull curve showed the best choice as a smoothing curve to represent the Kaplan–

Meier estimator's distribution for all studied fluids. Statistical analysis was conducted using R, version 3.6 (R Core Team, 2020) and IBM SPSS Statistics 22.0.

## Results

### Participants characteristics

From April 10^th to December 5^th, 2019, A total of 170 participants were screened. Of these, 152 patients were enrolled in the study. The reasons for exclusion of the 18 potential participants were as follows: unspecified viral disease (n = 13), bacterial tonsillitis (n = 1), syphilis (n = 1), mononucleosis (n = 1), influenza virus (n = 1), and adverse cutaneous reaction (n = 1), as shown in Fig 1.

The median and interquartile range (IQR) age was 39.3 (IQR; 26.9, 50.7) for men and 43.5 years (IQR; 33.8, 53.6) for women. The majority of the study population was women (n = 100, 65.8%). Most patients were born in the state of Rio de Janeiro (82.6%). Table 1 shows the main sociodemographic characteristics of the study population.

### Signs, symptoms, and comorbidities

Arthralgia (99.3%), fever (99.3%), prostration (94.7%), headache (86.8%), taste alteration (81.6%), chills (76.3%), myalgia (71.7%), and retroorbital pain (53.3%) were the most common symptoms reported. Complaints were associated with a rash in 84.9% of these patients. Joint swelling was also a common sign (61.2%), especially in the hands, ankles, and knees. Table 2 shows the signs and symptoms in the acute phase of the disease.

The most frequent comorbidities observed were high blood pressure (19.7%), allergic rhinitis (19.1%), and arthrosis (9.9%). Coinfection with HIV was found in eight patients (5.3%), but their clinical presentations did not differ from those of the rest of the study population.

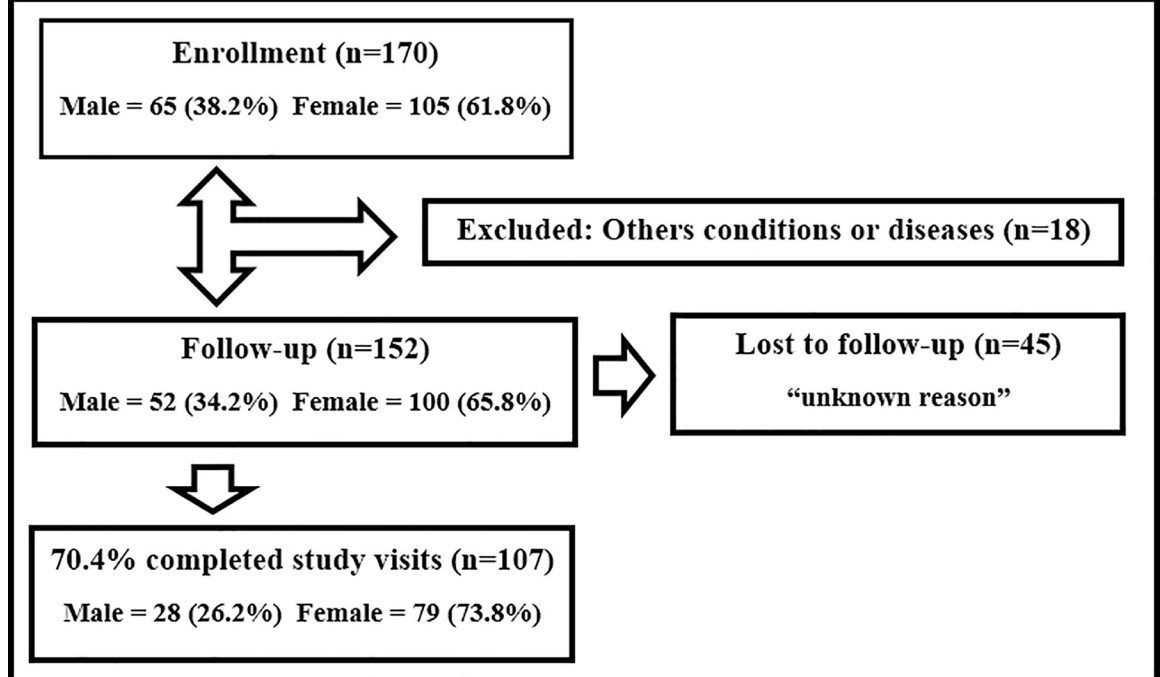

**Fig 1. Flow diagram of INOVACHIK Cohort Study.**

**Table 1. Sociodemographic characteristics of the study population, April—December 2019, Rio de Janeiro, Brazil.**

| Characteristics | n | % |
|---|---|---|
| **Female** | 100 | 65.8% |
| **Male median age (IQR)** | 39.3 (26.9–50.7) | |
| **Female median age (IQR)** | 43.5 (33.8–53.6) | |
| **Race** | | |
| Black | 24 | 15.8% |
| White | 76 | 50.0% |
| Mixed race | 51 | 33.6% |
| Yellow | 1 | 0.7% |
| **Education level** | | |
| Elementary School | 44 | 29.0% |
| High School | 66 | 43.4% |
| College | 42 | 27.6% |
| **Marital Status** | | |
| Single | 70 | 46.1% |
| Married / Stable Union | 66 | 43.4% |
| Divorced / Separated | 13 | 8.6% |
| Widowed | 3 | 2.0% |

IQR: interquartile range.

## Zika and Dengue virus coinfection

Among the 152 enrolled participants, three (2.0%) had confirmed ZIKV infection, as assessed by rRT-PCR in the serum (n = 2) and urine (n = 1). A positive rRT-PCR result for DENV was found in four participants (2.6%), and all presented with DENV-2 infection.

## Specific IgM antibodies against CHIKV

Specific IgM antibodies against CHIKV were detected in 146 participants (96.1%). Six participants had a single visit with negative IgM antibodies against CHIKV; therefore, it was not possible to document the occurrence of seroconversion. The lack of IgM class antibody production in these participants was probably because sample collection was performed 2 days after symptom onset in five participants and 3 days after symptom onset in another participant.

## CHIKV detection in bodily fluids

Among the enrolled participants, 122 had detectable CHIKV RNA (80.3%) in serum in at least one specimen (Table 3), and eight (5.3%) had CHIKV RNA detection in more than one serum sample. CHIKV RNA was detected in urine samples only once in 35 (23.0%) of 152 participants (Table 3). CHIKV RNA was detected in urine samples from 30/100 (30.0%) female participants and only 5/52 (9.6%) male participants. Among the 152 enrolled participants, 46 (30.3%) had CHIKV RNA detected in at least one saliva specimen (Table 3), four (2.6%) of whom had positive results more than once. Of the 52 male participants, 42 provided at least one semen sample. Only six participants (14.3%) had detectable CHIKV RNA in semen, of which two had a second detection. All but one female participant provided at least one vaginal secretion specimen for RT-PCR analysis. CHIKV RNA was present in 20 participants (20.2%), with a single detection performed in 19 participants and twice in one participant. Of note, six

**Table 2. Signs or Symptoms at Acute Phase of the Disease (1–14 days).**

| Sign/Symptoms | (n) | Percentage |
|---|---|---|
| Fever | 151 | 99.3% |
| Arthralgia | 151 | 99.3% |
| Prostration | 144 | 94.7% |
| Headache | 132 | 86.8% |
| Rash | 129 | 84.9% |
| Taste alteration | 124 | 81.6% |
| Chills | 116 | 76.3% |
| Pruritis | 115 | 75.7% |
| Anorexia | 114 | 75.0% |
| Myalgia | 109 | 71.7% |
| Backache | 100 | 65.8% |
| Nausea | 95 | 62.5% |
| Edema | 93 | 61.2% |
| Eye pain | 81 | 53.3% |
| Photophobia | 70 | 46.1% |
| Sweating | 50 | 32.9% |
| Diarrhea | 47 | 30.9% |
| Abdominal Pain | 46 | 30.3% |
| Lymphadenopathy | 36 | 17.1% |
| Eye Congestion | 30 | 19.7% |
| Dyspnea | 28 | 18.4% |
| Vomiting | 26 | 17.1% |
| Light Bleeding | 26 | 17.1% |
| Odynophagy | 25 | 16.4% |
| Runny nose | 23 | 15.1% |
| Nasal congestion | 21 | 13.8% |
| Otalgia | 18 | 11.8% |
| Cough | 16 | 10.5% |
| Hoarseness | 14 | 9.2% |
| Dysuria | 9 | 5.9% |

participants had their first CHIKV RNA detection at or after the third study visit (≥1 month after symptom onset).

Of the 152 enrolled participants, 114 were included in the persistence analysis. The reasons for exclusion were as follows: only reactive anti-Chikungunya IgM during the whole study

**Table 3. Detection of CHIKV RNA in Body Fluids, According to Gender for 152 enrolled participants\*.**

| Body Fluid | Total Patients | Positive Patients | Detection Percentage | Male n (%) | Female n (%) |
|---|---|---|---|---|---|
| Serum | 152 | 122 | 80.3% | 45 (86.5) | 77 (77.0) |
| Urine | 152 | 35 | 23.0% | 5 (9.6) | 30 (30.0) |
| Saliva | 152 | 46 | 30.0% | 14 (26.9) | 32 (32.0) |
| Semen | 42 | 6 | 14.3% | 6 (14.3) | Not Applicable |
| Vaginal Secretions | 99 | 20 | 20.2% | Not Applicable | 20 (20.2) |

\* Data were derived from a combination of all study visits. The first samples (for all fluids) were collected within the first week after symptom onset. Samples were collected every 15 days for 2 months and at 3 months of follow-up. Each participant collected a maximum of six samples for each body fluid.

**SERUM**

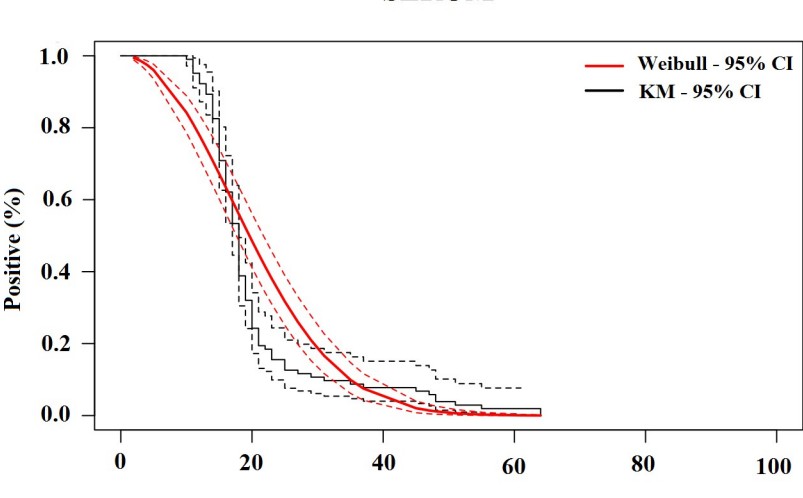

**Fig 2. Survival curves (Kaplan–Meier analysis) of CHIKV RNA persistence in Serum, with Weibull fit.**

(n = 25) without a positive RT-PCR result, the first detection of CHIKV RNA in any bodily fluids occurred during or after the third study visit (n = 6), coinfection with DENV-2 (n = 4), and ZIKV (n = 3). A total of 34 (29.8%) participants were lost to follow-up for unknown reasons, and 80 (70.2%) completed the study for persistence analysis.

The median time for the loss of CHIKV RNA detection was 19.6 days (95% CI, 17.5–21.7) in the serum (Fig 2), 25.3 days (95% CI, 17.8–32.8) in urine (Fig 3), 23.1 days (95% CI, 17.9–28.4) in saliva (Fig 4) and 25.8 days (95% CI, 20.6–31.1) in vaginal secretions (Fig 5). The number of semen samples available was too small for statistical estimation. Nevertheless, the maximum detection of CHIKV RNA was observed 56 days after the onset of symptoms in a study participant.

**URINE**

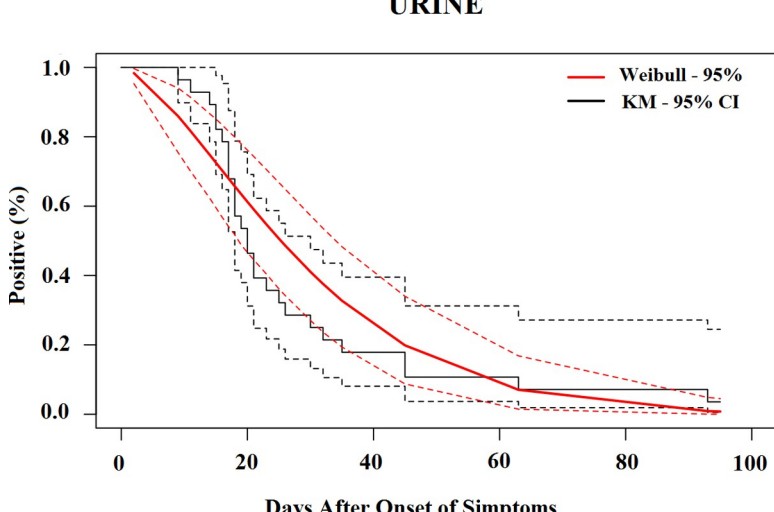

**Fig 3. Survival curves (Kaplan–Meier analysis) of CHIKV RNA persistence in Urine, with Weibull fit.**

**SALIVA**

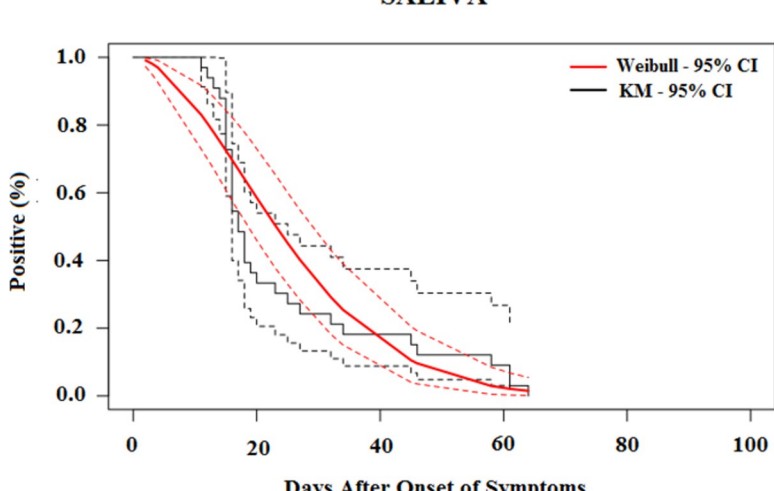

**Fig 4. Survival curves (Kaplan–Meier analysis) of CHIKV RNA persistence in Saliva, with Weibull fit.**

Table 4 shows the percentiles from the Weibull models and their 95% CIs until the loss of CHIKV RNA in selected body fluids. The 95th percentile of time was 39.7 days (95% CI, 35.7 to 43.7) in serum, 68.4 days (95% CI, 49.2 to 87.7) in saliva, 52.9 days (95% CI, 41.8 to 63.9) in urine, and 48.8 days (95% CI, 37.7 to 59.8) in vaginal secretions based on the Weibull model.

Figs 6–9 shows RT-PCR Ct values for CHIKV detection for each fluid studied, allowing the understanding of the strength of the signal in the different body fluids over time. Samples with a Ct < 38 (dashed line) were considered positive. Each circle indicates a positive result. As expected, Ct values were lower for all body fluids mainly in serum samples in the acute phase suggesting higher viral loads.

Real-time reverse transcription PCR cycle threshold values for Chikungunya virus. Samples with Ct <38 (dashed line) were considered to be positive. Each circle indicates a positive result.

**VAGINAL SECRETION**

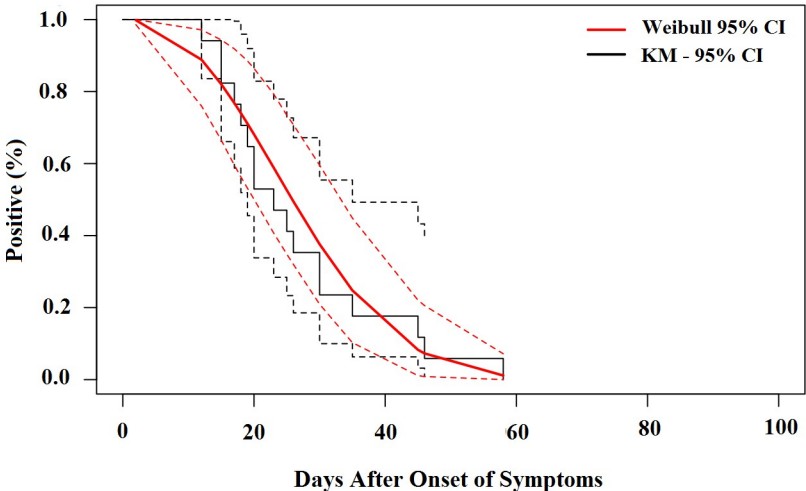

**Fig 5. Survival curves (Kaplan–Meier analysis) of CHIKV RNA persistence in Vaginal Secretion, with Weibull fit.**

**Table 4. Percentiles from the Weibull models until loss of CHIKV RNA in body fluids (in days).**

| Body Fluid | Percentile | 0.95 LCL | 0.95 UCL |
|---|---|---|---|
| | **25th** | | |
| Serum | 12.83 | 10.98 | 14.68 |
| Saliva | 13.93 | 8.37 | 19.49 |
| Urine | 14.09 | 9.78 | 18.39 |
| Vaginal Secretions | 17.65 | 13.34 | 21.96 |
| | **Median** | | |
| Serum | 19.60 | 17.47 | 21.73 |
| Saliva | 25.31 | 17.83 | 32.80 |
| Urine | 23.14 | 17.91 | 28.37 |
| Vaginal Secretions | 25.84 | 20.61 | 31.07 |
| | **75th** | | |
| Serum | 27.38 | 24.79 | 29.96 |
| Saliva | 40.53 | 30.11 | 50.96 |
| Urine | 34.22 | 27.58 | 40.86 |
| Vaginal Secretions | 34.90 | 28.26 | 41.54 |
| | **95th** | | |
| Serum | 39.69 | 35.73 | 43.65 |
| Saliva | 68.42 | 49.18 | 87.66 |
| Urine | 52.87 | 41.83 | 63.91 |
| Vaginal Secretions | 48.75 | 37.71 | 59.78 |

LCL: Lower Confidence Limit, UCL: Upper Confidence Limit

## Discussion

This longitudinal study reported CHIKV detection by RT-PCR in several bodily fluids, including genital secretions, during the acute and convalescent phases of the disease (up to 3

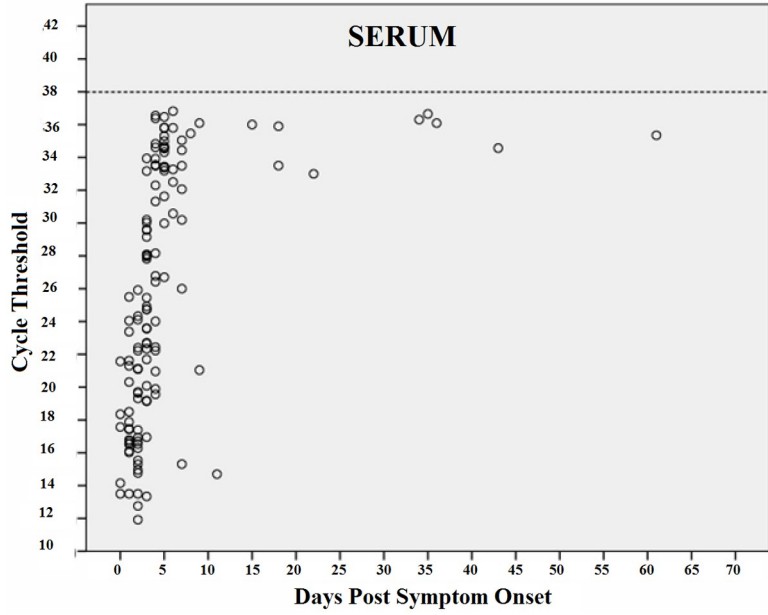

**Fig 6. CHIKV Cycle Threshold (Ct) by days after the onset of symptoms in Serum.**

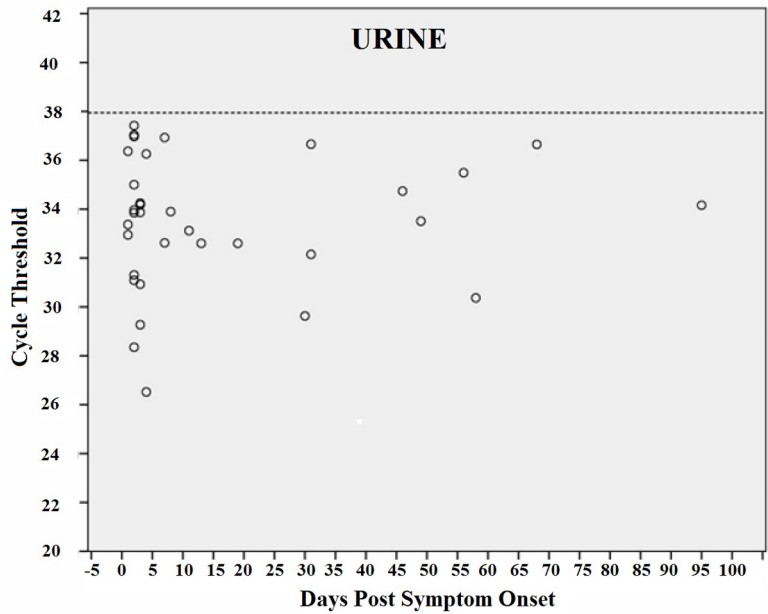

**Fig 7. CHIKV Cycle Threshold (Ct) by days after the onset of symptoms in Urine.**

months). We demonstrated that CHIKV RNA was detected more than 30 days in all the fluids studied. In addition, serum, urine, and saliva had detectable virus levels and persistence for more than 60 days, while urine had them for more than 90 days. To the best of our knowledge,

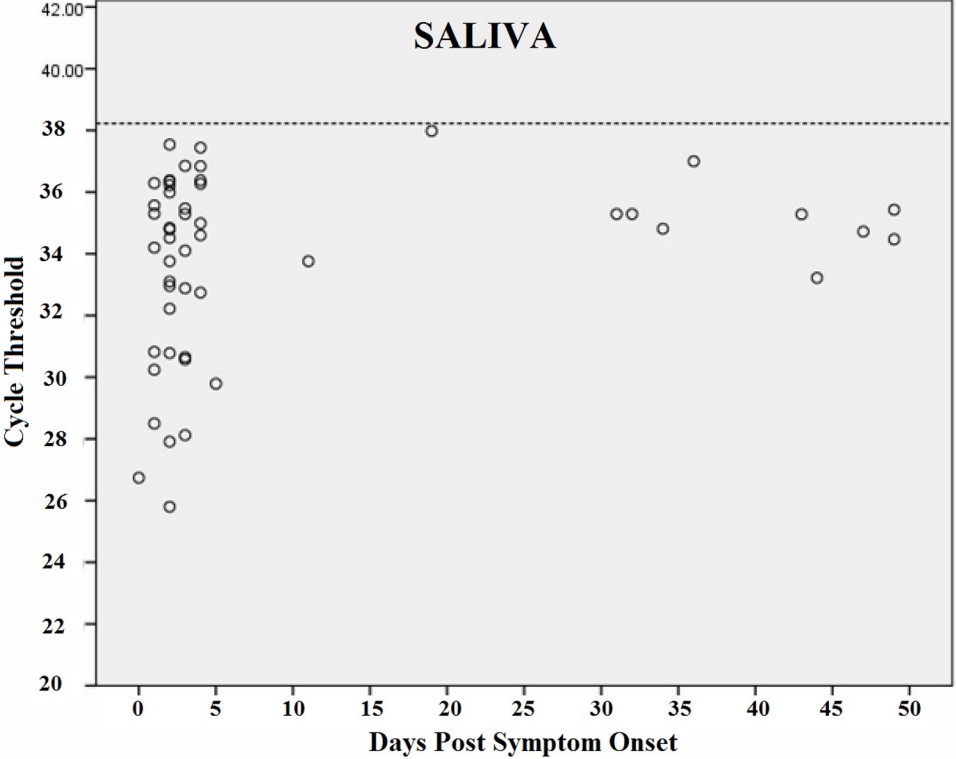

**Fig 8. CHIKV Cycle Threshold (Ct) by days after the onset of symptoms in Saliva.**

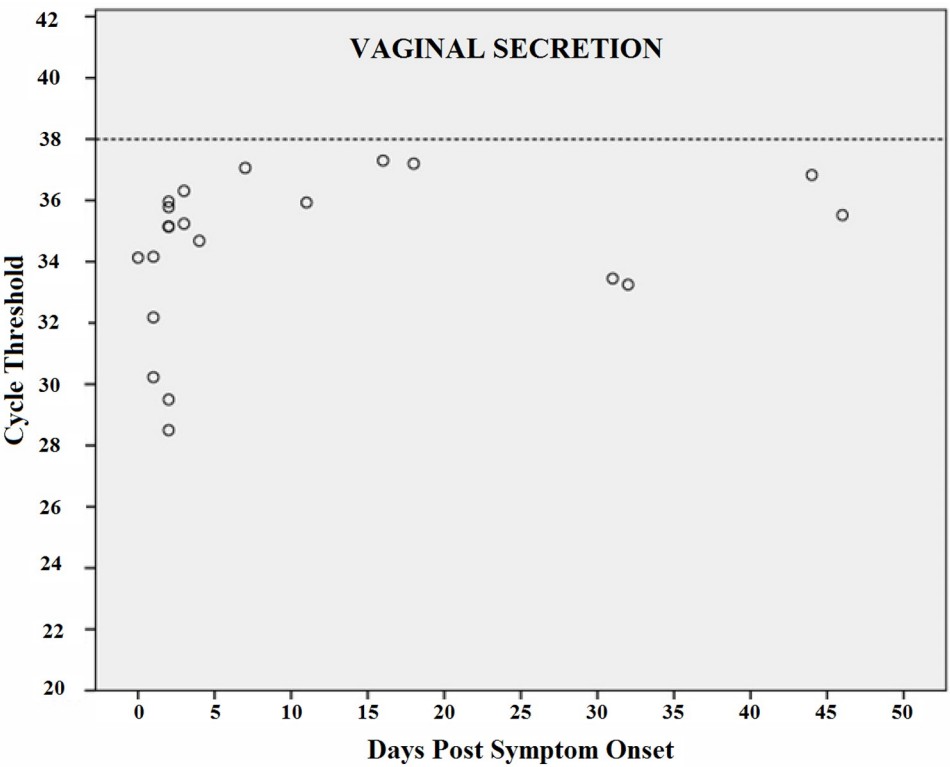

**Fig 9. CHIKV Cycle Threshold (Ct) by days after the onset of symptoms in Vaginal Secretion.**

this was the first cohort study to assess the persistence of CHIKV RNA in genital fluids (vaginal secretions and semen).

Females outnumbered male participants in the diagnosis of chikungunya. Similar results have been described in other studies [28–30]. A combination of fever, arthralgia, and prostration was the most prevalent presentation in our cohort, which is consistent with the results described by Anwar et al. [29].

Since 2014, the presence of co-circulating arboviruses (dengue, zika, and chikungunya) has increased the chance of coinfection. Epidemiological findings from a surveillance study for acute febrile illnesses including 948 participants, showed that 247 (26.1%) had evidence of an acute arboviral infection, of which 224 (23.6%) were single infections and 23 (2.4%) were coinfections [31]. Specifically, 13 (1.4%) patients tested positive for DENV/CHIKV coinfection and nine (0.9%) for CHIKV/flavivirus coinfection [28]. In another study, Dos Santos *et al*. reported five (9.6%) patients with coinfection with DENV-2 among 52 participants diagnosed with chikungunya [32]. Our cohort had similar results where coinfection of ZIKV and CHIKV was reported in three (2.0%) participants and 14 patients (9.2%) had reactive acute-phase anti-DENV IgM. DENV-2 was detected in only four participants (2.6%). We did not observe differences in symptom severity in patients with these coinfections.

We observed that the detection rate of CHIKV RNA was significantly higher in blood, saliva, and urine during the first week of symptom onset, which is consistent with other studies reporting viral presence during the acute phase of the disease [19,20]. In addition, saliva and urine did not increase the detection rate of CHIKV RNA in the acute phase of the disease, and, in concordance with Musso et al., blood was the sample of choice for chikungunya diagnosis [19]. CHIKV RNA persistence in the serum in our study was longer than expected. Most

literature reports showed that CHIKV RNA in serum declines to undetectable levels within 1–2 weeks after symptom onset [33–35].

We also detected CHIKV RNA in urine 95 days after symptom onset. A similar study by Bandeira *et al*. reported the maximum viral persistence in urine after 30 days [20]. Interestingly, in our cohort, CHIKV RNA was detected in 30% of urine samples from female participants and in only 9.6% of male participants. We did not find reports evaluating RT-PCR RNA detection rates in urine samples by sex, but contamination by menstrual blood can be a reasonable explanation, although all the guidelines for urine collection were given to the study participants. Additionally, the collection was not performed during the menstrual period.

To the best of our knowledge, this is the first prospective study to monitor and detect CHIKV RNA in vaginal secretions. We detected CHIK RNA up to 46 days after the acute onset of symptoms in vaginal secretion samples. We did not perform statistical estimates for semen as the number of samples was small, but the maximum detection of CHIKV RNA in semen was 56 days after the onset of symptoms in a study participant.

Although we detected CHIKV in semen and vaginal secretions, it was impossible to assess its potential for sexual transmission as viral isolation was not attempted. In addition, this study did have an appropriate study design to establish sexual transmission due to the endemic nature of the infection, making it difficult to ascertain the actual route of transmission, sexual or vectorial, but additional studies on viral infectivity are warranted. Therefore, it was out of the scope of this study to assess the sexual transmission of CHIKV.

Chikungunya diagnosis in humans is mainly based on RNA detection in serum or plasma samples. However, we have demonstrated that saliva and urine could be considered as potential alternative samples for diagnosis in the acute and convalescent phases of the disease. Diagnostic algorithms using urine or saliva as alternative samples have the advantage of being quick, easy-to-perform, and being less invasive than blood collection. The demonstration of longer persistence of CHIKV in bodily fluids may help diagnosis in later stages of the disease.

This study has some limitations. 1) As the visits and sample collections were scheduled every 15 days, we may have underestimated the exact viral persistence time in the different body fluids; 2) the median duration of CHIKV in semen was evaluated in a small number of patients, because of difficulties in sample collection, mainly due to joint pain in the acute phase of the disease; and 3) the follow-up time was limited to 90 days, making it impossible to assess the maximum persistence of CHIKV in all bodily fluids.

Knowledge of chikungunya viral persistence, infectivity and epidemiology can inform recommendations for control, treatment, and prevention of the disease, and contribute to public health programs.

## Acknowledgments

The authors thank the patients for participating in this study and all employees of the Oswaldo Cruz Foundation.

## Author Contributions

**Conceptualization:** Ezequias B. Martins, Wagner S. Tassinari, Fernanda de Bruycker-Nogueira, Ana Maria B. Filippis, Guilherme A. Calvet.

**Data curation:** Ana Maria B. Filippis, Guilherme A. Calvet.

**Formal analysis:** Ezequias B. Martins, Wagner S. Tassinari, Guilherme A. Calvet.

**Funding acquisition:** Ana Maria B. Filippis, Guilherme A. Calvet.

**Investigation:** Ezequias B. Martins, Michele F. B. Silva, Fernanda de Bruycker-Nogueira, Isabella C. V. Moraes, Cintia D. S. Rodrigues, Carolina C. Santos, Simone A. Sampaio, Anielle Pina-Costa, Allison A. Fabri, Vinícius Guerra-Campos, Nayara A. Santos, Nieli R. C. Faria, Ana Maria B. Filippis, Patrícia Brasil, Guilherme A. Calvet.

**Methodology:** Ezequias B. Martins, Wagner S. Tassinari, Guilherme A. Calvet.

**Project administration:** Ana Maria B. Filippis, Guilherme A. Calvet.

**Resources:** Ezequias B. Martins, Michele F. B. Silva, Isabella C. V. Moraes, Anielle Pina-Costa, Ana Maria B. Filippis, Patrícia Brasil, Guilherme A. Calvet.

**Software:** Wagner S. Tassinari, Guilherme A. Calvet.

**Supervision:** Ana Maria B. Filippis, Guilherme A. Calvet.

**Validation:** Ezequias B. Martins, Wagner S. Tassinari, Fernanda de Bruycker-Nogueira, Cintia D. S. Rodrigues, Carolina C. Santos, Simone A. Sampaio, Allison A. Fabri, Vinícius Guerra-Campos, Nieli R. C. Faria, Ana Maria B. Filippis, Guilherme A. Calvet.

**Visualization:** Ezequias B. Martins, Wagner S. Tassinari, Guilherme A. Calvet.

**Writing – original draft:** Ezequias B. Martins, Wagner S. Tassinari, Patrícia Brasil, Guilherme A. Calvet.

**Writing – review & editing:** Ezequias B. Martins, Michele F. B. Silva, Wagner S. Tassinari, Fernanda de Bruycker-Nogueira, Isabella C. V. Moraes, Cintia D. S. Rodrigues, Carolina C. Santos, Simone A. Sampaio, Anielle Pina-Costa, Allison A. Fabri, Vinícius Guerra-Campos, Nayara A. Santos, Nieli R. C. Faria, Ana Maria B. Filippis, Patrícia Brasil, Guilherme A. Calvet.

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
