## [Decision Letter · Decision Letter 0]

12 Nov 2021

Dear Dr Martins,

Thank you very much for submitting your manuscript "Detection of Chikungunya Virus in bodily fluids: The INOVACHIK Cohort Study." for consideration at PLOS Neglected Tropical Diseases. As with all papers reviewed by the journal, your manuscript was reviewed by members of the editorial board and by several independent reviewers. In light of the reviews (below this email), we would like to invite the resubmission of a significantly-revised version that takes into account the reviewers' comments. 

We cannot make any decision about publication until we have seen the revised manuscript and your response to the reviewers' comments. Your revised manuscript is also likely to be sent to reviewers for further evaluation.

Sincerely,

William B Messer

Associate Editor

Benjamin Althouse

Deputy Editor

Reviewer's Responses to Questions

**Key Review Criteria Required for Acceptance?**

**Methods**

-Are the objectives of the study clearly articulated with a clear testable hypothesis stated?

-Is the study design appropriate to address the stated objectives?

-Is the population clearly described and appropriate for the hypothesis being tested?

-Is the sample size sufficient to ensure adequate power to address the hypothesis being tested?

-Were correct statistical analysis used to support conclusions?

-Are there concerns about ethical or regulatory requirements being met?

Reviewer #1: See below

Reviewer #2: The methods are generally well-described, but some points should be clarified:

A description of how the RNA was extracted (Trizol, RLT, AVL, etc), cDNA generation, and if any post-PCR analysis was performed is missing.

Line 111: what do the authors mean by "focus of infection"? Please clarify the specific criteria for the time post symptom onset and positive CHIKV PCR to the time of enrollment in the study?

Line 141-142: What is the rationale or experimental data for the assumption that all the fluids would be positive at symptom onset?.

Line 144-145: What is the rationale for censoring these individuals? If they are still positive at the end of the study, this would still indicate CHIKV RNA persistence, supporting the other data that CHIKV RNA persists long-term.

Table 1: Race/ethnicity data: are these the correct terminologies for different race and ethnicities?

**Results**

-Does the analysis presented match the analysis plan?

-Are the results clearly and completely presented?

-Are the figures (Tables, Images) of sufficient quality for clarity?

Reviewer #1: See below

Reviewer #2: The results are clearly presented in the figures and tables. Some of the data are not shown, and the paper would benefit from the following added:

Line 166-175: Please report the signs and symptoms in a table.

Line 181-184: Please report the IgM ELISA titers in a figure.

Figure 2: The median and maximum time to a negative CHIKV RNA result is reported in the text, but it would be useful to have these numbers also represented in the figure or in a separate table.

**Conclusions**

-Are the conclusions supported by the data presented?

-Are the limitations of analysis clearly described?

-Do the authors discuss how these data can be helpful to advance our understanding of the topic under study?

-Is public health relevance addressed?

Reviewer #1: See below

Reviewer #2: The introduction and conclusions/discussion could be reworked to highlight the public health significance of the work. For example, in the discussion, the authors could discuss how the urine or saliva test is less-invasive compared to blood tests for viral RNA. The downside is that the viral RNA is most consistently detected in the serum (80% positivity in this dataset). The authors could also expand upon the difference in detecting viral RNA compared to isolating infectious virus and any implications that this might have.

**Editorial and Data Presentation Modifications?**

Reviewer #1: (No Response)

Reviewer #2: Lines 26-27: Needs a little more information in the abstract background section. A statement that introduces CHIKV is missing. A statement is needed about what was done and why the detection of CHIKV in body fluids may be important for public health.

Line 52: "...are warranted."

Line 52: Expand on the conclusion section- what are the implications for long-term detection of CHIKV RNA in body fluids?

Line 67: "...warranted."

Line 69: State where CHIKV is geographically distributed.

Line 73: CHIKV disease is generally described in three phases: acute, post-acute, and chronic. An expansion of the clinical symptoms of each phase is needed.

Line 83: There should be a more complete description of what has been shown for CHIKV detection from body fluids over each phase of infection. Also add the citation with Joy Gardner, et al. 2015 Plos One that showed infectious CHIKV recovery from patient saliva and compared it to animal models.

Line 83: Remove the word "other". Is anything known for other alphaviruses (RRV, ONNV, MAYV, VEEV, etc) in terms of detection in other body fluids such as saliva? 

Line 183-184: In reference to the six participants that had a single visit with negative IgM antibodies- what was their time post symptom onset?

Line 271-276: Expand this paragraph to discussing CHIKV transmission as a whole. Please include a statement about how transmission occurs primarily through mosquitos but we do not know if transmission may occur by saliva or sexual contact (as with ZIKV for example).

**Summary and General Comments**

Reviewer #1: In this manuscript, Martins et al. used RT-PCR to evaluate body fluids, collected at enrollment and out to three months from Brazilian patients with confirmed chikungunya virus (CHIKV) infection, for the presence of CHIKV RNA. CHIKV RNA was detected in all sample types (e.g., serum, urine, saliva, semen, vaginal secretions) at different frequencies and for different lengths of time since enrollment. A major strength of the study is the remarkable paucity of this type of virological data for CHIKV infection in humans. A major limitation of the study is the absence of synovial fluid, joint-associated tissues or other musculoskeletal tissues. Nevertheless, the study contributes important virological information. However, some of the presentation of the data is confusing and/or incomplete. Specific comments to improve the manuscript are outlined below. 

1. The presentation of the data in Table 2 is confusing. Is this data derived from the initial sample collection at the time of enrollment? Or is it combined data from multiple collections? In some cases, the text says “in at least one specimen” or other similar statements. The presentation of these data needs to be clarified and improved. In addition , it also seems important to include information about the time of disease onset relative to collection of the initial samples. 

2. Which collection times were used for the persistence analysis and which ones were used for the acute analysis (is this Table 2?). This should be clarified. 

3. The details of RNA isolation, cDNA generation, and the RT-PCR assay are missing and should be included in the materials and methods section.

4. The authors state that any reaction with a Ct value lower than 38 and with sigmoid curves was considered positive. It would improve the study to include the Ct values detected to give readers an opportunity to understand the strength of the signal in the different body fluids over time. 

5. Given the sensitivity of RT-PCR and the potential for false positives, the authors should report the methods and measures used to minimize contamination and other possible sources of false positive signals. 

6. The clinical data presented from lines 166-175: Are these the signs and symptoms that were present at the time of study enrollment and first sample collection? This is not clear.

7. The authors indicate that clinical data were collected throughout the study, however, these are not reported. It would be useful to compare the clinical presentation to the decline in viral RNA signal in samples. Is there any relationship to the resolution and/or persistence of clinical symptoms and signs?

Reviewer #2: In this work, the authors describe a clinical study where they sampled body fluids from CHIKV patients (serum, saliva, urine, semen, and vaginal secretions) throughout the post-acute phase of disease. No previous group has examined viral RNA persistence in over time in all these body fluids, and no group has examined viral RNA in semen and vaginal secretions. The authors found that CHIKV RNA generally persists in body fluids for about 20-25 days. However, viral RNA detection in the serum was still the most reliable readout, with most (80%) of the cohort having CHIKV RNA positivity in the serum over the post-acute phase. The study would benefit from a discussion of the implications of these data for public health. It would also benefit from additional reasoning for why they were looking for viral RNA in body fluids other than serum, and more organization and flow in the introduction and conclusions section.

The following points should be addressed:

1. In the introduction, there should be a more detailed discussion of what we know about CHIKV RNA detection in patient body fluids (serum, urine, saliva, etc) during each infection phase- the acute, post-acute phase, and chronic phase. In addition, CHIKV detection in body fluids from CHIKV animal models could enhance the discussion and provide some comparison.

2. The long-term detection of vRNA shed in bodily fluids could reflect either infectious virus in the bodily fluids, or vRNA in the absence of infectious virus. The genome copy number would be useful because high copy numbers may be more suggestive that infectious virus is present. The authors should report CHIKV levels as genome copies (or Ct's if they don't have genome copies) relative to days after symptom onset. In addition, please describe the detection limit of the PCR assay. 

3. In Figure 2- I'm not an expert on statistics, but the Weibull curve diverges from the KM curve in the serum, urine, and saliva data, but not as much in the vaginal secretion data. Why is this the case? Are both statistical analysis needed?

4. The conclusions section could use more discussion of the significance of CHIKV detection in body fluids. It would also benefit from a more organized structure. For example, there could be a paragraph devoted to CHIKV RNA detection in acute phase and a separate paragraph on long-term detection.

PLOS authors have the option to publish the peer review history of their article (what does this mean?). If published, this will include your full peer review and any attached files.

Reviewer #1: No

Reviewer #2: No
---

## [Decision Letter · Decision Letter 1]

9 Feb 2022

Dear Dr Martins,

We are pleased to inform you that your manuscript 'Detection of Chikungunya virus in bodily fluids: The INOVACHIK Cohort Study.' has been provisionally accepted for publication in PLOS Neglected Tropical Diseases.

Best regards,

William B Messer

Associate Editor

Benjamin Althouse

Deputy Editor

Reviewer's Responses to Questions

**Key Review Criteria Required for Acceptance?**

**Methods**

-Are the objectives of the study clearly articulated with a clear testable hypothesis stated?

-Is the study design appropriate to address the stated objectives?

-Is the population clearly described and appropriate for the hypothesis being tested?

-Is the sample size sufficient to ensure adequate power to address the hypothesis being tested?

-Were correct statistical analysis used to support conclusions?

-Are there concerns about ethical or regulatory requirements being met?

Reviewer #1: Please see summary

**Results**

-Does the analysis presented match the analysis plan?

-Are the results clearly and completely presented?

-Are the figures (Tables, Images) of sufficient quality for clarity?

Reviewer #1: Please see summary

**Conclusions**

-Are the conclusions supported by the data presented?

-Are the limitations of analysis clearly described?

-Do the authors discuss how these data can be helpful to advance our understanding of the topic under study?

-Is public health relevance addressed?

Reviewer #1: Please see summary

**Editorial and Data Presentation Modifications?**

Reviewer #1: Please see summary

**Summary and General Comments**

Reviewer #1: The authors have addressed the comments from the prior review and the manuscript is substantially improved. The inclusion of Ct values in new Fig 3, clarification of sample collection and symptom assessment, inclusion of assay details and methods to prevent contamination, and other modifications have greatly strengthened the report.

PLOS authors have the option to publish the peer review history of their article (what does this mean?). If published, this will include your full peer review and any attached files.

Reviewer #1: No

---

## [Editor Report · Acceptance letter]

17 Feb 2022

Dear Dr Martins,

We are delighted to inform you that your manuscript, "Detection of Chikungunya virus in bodily fluids: The INOVACHIK Cohort Study.," has been formally accepted for publication in PLOS Neglected Tropical Diseases.

Best regards,

Shaden Kamhawi

co-Editor-in-Chief

Paul Brindley

co-Editor-in-Chief
